# Immunomodulation and Immunotherapy for Patients with Prostate Cancer: An Up-to-Date Review

**DOI:** 10.3390/biomedicines13051179

**Published:** 2025-05-12

**Authors:** Nigel P. Murray

**Affiliations:** 1Faculty of Medicine, Universidad Finis Terrae, Santiago 7501015, Chile; nigelpetermurray@gmail.com; 2Department of Medicine, Hospital de Carabineros de Chile, Santiago 7770199, Chile

**Keywords:** prostate cancer, immunotherapy, cancer vaccines, biochemical failure, metastasis

## Abstract

Immunotherapy alone or in combination with chemotherapy or radiotherapy is the frontline treatment for melanoma and lung cancer. However, its role in prostate cancer is usually as a fourth-line treatment. It is usually employed in patients with metastasis, after androgen blockade and chemotherapy. This article reviews the immunosuppressive effects of prostate cancer and possible uses of various types of immunotherapies. It also considers when would be the optimal time to employ this type of therapy.

## 1. Introduction

In 2022, of all the men diagnosed with cancer, 14% of these had prostate cancer [1], while in American men this figure accounted for 19% of all cancers, representing 9% of all cancer deaths [2]. However, for those patients with localised cancer treated with curative radical prostatectomy or external beam radiotherapy or brachytherapy the 5-year survival rate is over 99% [3]. During follow-up, serial measurements of the serum PSA are made, to determine the absence or presence of biochemical recurrence. This has been defined as a serum PSA > 0.2 ng/mL after radical prostatectomy [4] or, according to the ASTRO II (American Society for Therapeutic Radiology and Oncology) guidelines, an increase in the serum PSA of 2 ng/mL above the nadir level achieved after radiotherapy has been defined as biochemical recurrence [5]. However, the prostate is a multifunctional organ; there are glandular cells that produce seminal fluid, ductal cells that line the ducts that allow the flow of seminal fluid, neuroendocrine cells which produce hormones such as testosterone and small cell carcinomas. Each subtype has different genotypic and phenotypic characteristics and thus differing biological properties which may influence treatment decisions. Most prostate cancers are adenocarcinomas arising from the glands that produce seminal fluid, approximately 90–95% of cases [6,7,8], rising to 99% in Chinese patients [7]. This subtype of prostate cancer is sensitive to androgen deprivation therapy (ADT).

Rarer subtypes of prostate cancer include intraductal and ductal adenocarcinoma; intraductal cancer has a layer of p63 positive basal cells whereas ductal carcinoma does not [9]. It accounts for 0.17% of all prostate cancers [10] and according to Cui et al. [11] there is no recognised best course of action to treat this subtype of prostate cancer and its prognosis is unclear now. It has been reported that this subtype of prostatic cancer is aggressive and is associated with a faster progression to lethal disease [12,13]. It has also been reported that the tumour microenvironment is more immunosuppressive when there are an increased number of cancer-associated fibroblasts, M2-type macrophages and more dysfunctional and fewer cytotoxic T-cells [10]. Intraductal cancer has been divided into two subgroups depending on the infiltration by immune cells: “cold” tumours have lower Tregs, CD163^+^ macrophages and CD83^+^ dendritic cells as compared to the adjacent adenocarcinoma cells. The denominated “hot” intraductal prostate cancer shows an inverse leukocytic pattern of infiltration and was associated with a higher risk dissemination and metastasis formation [14]. In 2025, a new classification of intraductal prostate cancer was suggested. 

This was based on the morphological characteristics of the tumour: a sieve-like or cribriform structure found in the specimen was deemed Pattern 1; while those cancers with a solid or dense cribriform morphology were classified as Pattern 2. Patients with Pattern 2 intraductal had an increased risk of biochemical failure and decreased overall survival when treated with radical prostatectomy as monotherapy [15]. In addition, unlike that found in prostatic adenocarcinoma, the location of metastasis is different, including penile and peritoneal metastasis [10]. 

The third subtype is that of neuroendocrine cancer: the prostate secretes hormones such as testosterone which is produced by the neuroendocrine cells. Typically, these cells do not express PSA or PMSA but are positive for the expression of synaptophysin, chromogranin A and CD56 [16]. De novo neuroendocrine prostate cancer is infrequent, comprising less than 2% of all prostate cancers [17]. It is an aggressive subtype of prostate cancer usually presenting as a metastatic disease with a limited survival time of approximately 17 months [18]. The second type of neuroendocrine cancer is in the context of androgen deprivation therapy—although most patients initially respond to androgen blockade there is a progression to castration-resistant disease. The progression to treatment-derived neuroendocrine prostate cancer changes the tumour from androgen dependent to an androgen-independent disease, resulting in a more aggressive disease and poorer prognosis [19]. It is frequently under-diagnosed, especially in the absence of diagnostic biomarkers, but up to 16% of patients treated with androgen deprivation therapy, especially with androgen receptor pathway inhibitors, will develop neuroendocrine prostate cancer as a late complication of androgen deprivation therapy [20]. This transformation due to trans-differentiation or linear reprogramming results in the loss of expression of PSA and the androgen receptor while there is up-regulation of neuron-specific enolase, chromogranin-A, synaptophysin and CD56, resulting in resistance to androgen deprivation therapy [21]. Paraneoplastic syndromes may be also found in neuroendocrine prostate cancer, SIADH, Cushing’s syndrome, dermatomyositis, polycythaemia and chronic intestinal dysmotility and was recently reviewed [19,22,23]. The optimum treatment of these cancers has not been established and chemotherapy with platinum-based agents with or without etoposide has been utilised with limited success [16]. Treatment with docetaxel, cabazitaxel and checkpoint inhibitors has also only had limited success [24]. 

Small cell prostate cancer has been classified as a subtype of neuroendocrine prostate cancer and, similarly to neuroendocrine prostate cancer, has an aggressive phenotype and more frequent visceral, lytic bone metastasis, a low or normal PSA and a lack of response to androgen deprivation therapy [25]. It typically develops after prolonged treatment, especially after androgen deprivation therapy and the increased use of androgen receptor signalling inhibitors such as abiraterone, darolutamide and apalutamide [26].

However, prostate cancer is complex, with diverse cell types within its onco-sphere; recently, scRNA-seq-profiling has shown that there is a substantial heterogeneity both the inter- and intratumoral of the different prostate cancer subtypes, both in untreated and castration-resistant metastatic prostate cancer [27]. This explains in part why there may be a transformation of subtypes to form a mixed tumour cell population, such as ductal adenocarcinoma having a phenotype of neuroendocrine prostate cancer [28], and why ductal and acinar components of mixed prostate cancer share a common clonal origin [29]. 

The presence of biochemical recurrence implies the dissemination of cancer cells to distant tissues before curative treatment. Moreno et al. reported the dissemination of prostate cancer cells first to the neurovascular structures and then into the circulation [30]. It has even been reported that circulating tumour cells can be detected before a prostate biopsy, highlighting their early dissemination into the circulation [31]. It has been estimated that approximately 10^6^ cancer cells per gram of the primary tumour are released into the circulation over a 24 h period [32]. However, the metastatic process is inefficient: the majority of these circulating tumour cells are destroyed by the sheer forces found in the circulation or by the innate and/or acquired immune systems with only an estimated 0.01% of these cells surviving [33]. Some of these cancer cells can implant in distant tissues and survive. They enter a varying latency period or dormancy before “awaking” to proliferate and grow.

To understand the rational of immunotherapy it is necessary to understand the complex interactions between the cancer cells, both in the primary tumour and distant micro-metastasis and the immune system.

## 2. Modulation of the Immune System by the Primary Tumour

Primary prostate cancer is made of different subpopulations of tumour cells. This heterogeneous population has differing genotypic and phenotypic characteristics and therefore differing biological properties [34]. Not only this, but the cancer cells reside in the tumour microenvironment (TME). This TME is composed of host cells, fibroblasts, immune cells, blood vessels, endothelial cells, and the extracellular matrix for example. Paget in 1891 proposed the hypothesis of the idea of the soil and seed; those seeds (cancer cells) which fell on stony ground would not germinate or proliferate, while those that fell on fertile ground would germinate, proliferate, and eventually cause metastatic disease [35,36]. The interactions between the different TME components are dynamic and change with time. This multi-factorial process has been described as an ecosystem, classifying it as a “multidimensional, spatiotemporal unity of ecology and evolution” [37]. In this ecosystem there is an intraspecific relationship—that is, communication between cells—and an interspecific relationship between the cancer cells and host factors. It has further been described that this ecosystem represents the total of the primary, regional, distal, and systemic “onco-spheres”, each with its own local microenvironment, niches and immune, nervous and endocrine systems [38].

The immune response can be basically split into two: firstly, the component which is responsible for the elimination of the cancer cells. This is composed of the innate immune response; whereby natural killer cells (NK-cells) play a cytotoxic role to eliminate tumour cells and the acquired immune response carried out by CD8 positive cytotoxic T-lymphocytes while dendritic cells present neo-antigens to the cytotoxic T-cells to enhance the immune response. On the other hand, there are immune cells which inhibit the effector cells. These include CD4, FOXP3 positive regulatory T-lymphocytes (Tregs), myeloid derived suppresser cells (MDSCs) recruited from the bone marrow and tumour-associated macrophages. The balance between cytotoxic cells and suppressor cells will determine if the tumour cells are eliminated or not. As previously mentioned, the tumour onco-sphere is complex and dynamic, with various mechanisms which affect this balance.

It is important that this ecosystem has multiple interactions but for the purpose of describing the role of each of factor it is necessary to limit this to the individual components.

**Prostate cancer cells:** Tumour cells can directly affect the immune system both cytotoxic and immunosuppressive systems. Using single-cell RNA sequencing and multicolour flow cytometry, it has been reported that high-grade localised prostate cancer is highly infiltrated with exhausted cytotoxic T-cells, which are expressed as TIM3, TOX, PD-1, CTLA4, CXCL13 and other markers of immune exhaustion in high levels, as well as MDSCs and Tregs. This is contrasted with the results found in low-grade prostate cancer, where the inverse was found in the ratio of cytotoxic T-cells relative to Tregs and MDSCs. In high-grade prostate cancer, tumour-infiltrating lymphocytes (TILs) expressed high levels of the androgen receptor and prostate-specific membrane antigen but less PSA antigen when compared to low-grade prostate cancer [39]. PTEN is a tumour suppressor gene; its loss has been reported to be in the order of 20–30% of newly diagnosed prostate cancers. The loss of the PTEN gene increases with disease progression and its inactivation has been reported to paradoxically cause immunosuppression [40]. Tregs were found to be significantly increased in PTEN-negative prostate cancer, but this was dependent on the site of sampling. In bone metastasis, it caused a decrease in cytotoxic T-cells, in liver metastasis it caused the Tregs to be increased, while in lymph nodes cytotoxic T-cells were increased. These different alterations in the immune function also highlight the importance of the TME, with PTEN loss causing different effects depending on the surrounding tissue.

The BRCA mutations cause defects in the reparation of damaged DNA. The location of immune cells in wild-type BRCA 1/2 tumours is predominately extra-tumoral, contrasting with an intra-tumoral location in mutated forms. The ratio of intra- to extra-tumoral locations of immune cells was significantly higher in mutated tumours, especially for CD4, CD8 and FOXP3 lymphocytes. However, the FOXP3 to cytotoxic T-cells ratio was significantly higher in BRCA-mutated cancer, implying a more immunosuppressed TME [41].

The extracellular matrix is a complex system of proteins, glycoproteins and proteoglycans, which form aggregates such as sheet-lime networks and fibrils. In addition, it has biophysical properties such as molecular density, rigidity and tensional forces [42].

Matrix metalloproteinase 2 (MMP-2) is a type IV collagenase; its expression in prostate cancer cells is associated with a worse prognosis [43,44]. Not all prostate cancer cells express MMP-2; those which do are able to pass through the basement membrane and the extracellular matrix and enter the circulation; cells that are MMP-2-negative may disseminate in a form of Indian file through the tract left by MMP-2-positive cells. The multiple interactions between the tumour and stromal cells causes a re-programming of the phenotypic characteristics of both components [43,44]. MMP-2, in degrading the extracellular matrix, triggers the proteolysis of cytokines and their respective receptors such as tumoral necrosis factor receptor R, interleukin 6R and 2R [45]. In addition, MMP-2 causes TH2 polarisation of macrophages from the TH1 subtype, therefore restricting the antitumour immune response. It is also able to cleave the interleukin-2-R-alpha receptor which, in doing so, suppresses the proliferation of cytotoxic T-cells because of increased apoptosis [46]. In addition to causing local immunosuppression, they also cause immunosuppression at distant sites by the release of exosomes into the circulation. These exosomes are released by a process of exocytosis, entering the circulation, and thus can reach distant tissues. Exosomes are membrane-bound vesicles with a diameter of approximately 50–100 nm; they fuse with the tumour cell plasma membrane and are thus able to disseminate into the circulation. Exosomes are thought to be important in both the communication between cancer cells and in the remodelling of the TME [47]. Due to the differential expression of integrins on their membrane, this may play in part their role in the organotropic distribution of metastasis [48]. The exosomes fuse directly with the membranes of target cells, releasing their contents into the normal cells occupying the TME [49]. The exosomal contents then cause immune dysfunction; firstly, they cause the differentiation of bone marrow monocytes into MDSCs, causing local immunosuppression, and these cells also migrate to the primary tumour, enhancing the immunosuppressive TME [50,51].

Secondly, they selectively affect T-cell maturation, decreasing the number and activity of cytotoxic T-cells [52], increasing the proliferation of CD4^+^ T-cells [53] and decreasing the cytotoxic capabilities of NK-cells [52] and conversion of immature B-lymphocytes into regulatory B-cells [53]. Finally, they increase the proliferation and activity of Tregs [54], the activity of MDSCs [51] and the immunosuppression caused by regulatory B-lymphocytes [53].

As a result of these changes, the immunosuppression of the pre-metastatic niche makes it fertile soil for future circulating tumour cells (the seeds) to implant in these distant sites.

Fibroblasts maintain the architecture of the TME. The definition of a fibroblast is somewhat difficult, mostly deriving from the embryonic mesoderm, although some derive from the neural crest. As they lack cell specific biomarkers, they are often defined by their morphology, localisation in the tissue and a lack of epithelial, mesenchymal and white cell markers (Figure 1).

## 3. The Role of Immunotherapy and Immunomodulation

Although the use of immunotherapy has changed the treatments used and of immunosuppression in renal cell and bladder cancer [55,56], this has not been the case in patients with prostate cancer. classify prostate cancer as immunologically “cold” with a TME that is essentially immunosuppressive [57]. This immunological TME changes with time, the primary tumour being “colder” than metastatic disease [58]. In metastatic bone disease, the immunosuppression mechanisms and lack of cytotoxic cell activation result in a TME immunosuppressant environment and thus immunotherapy is much less effective than in other solid tumours [59]. The immunological characteristics of the TME change with time or due to treatment [60]. Therefore, the identification of biomarkers that characterise the molecular, phenotypic and biological properties that may predict the benefit of immunotherapy are required [61]. Most reports of the activity of immunotherapy have been in patients with castration-resistant metastatic prostate cancer, who have relapsed after first- and second-line anti-androgens and/or chemotherapy with a taxane. This is in contrast with the early use of immunotherapy in malignancies, such as the use of trastuzumab in HER-2-positive breast cancer and the anti-CD20 monoclonal antibody rituximab in non-Hodgkin’s lymphoma. Secondly, it is possible in patients with macro-metastatic disease which, as revealed by imaging studies, may not permit the entry of large molecules such as antibodies, or that the TME is sufficiently immunosuppressive that cytotoxic immune cells are not able to enter the metastasis to eliminate the tumour cells.

**Sipuleucel-T** is an autologous cellular immunotherapy used in patients with metastatic castration-resistant prostate cancer. These patients may be asymptomatic or with a minimum of symptoms. Sipuleucel-T was shown to reduce the risk of death and increase overall survival as compared with a placebo in the Immunotherapy for prostate adenocarcinoma treatment phase 3 trial (IMPACT) in 2010 [62]. Based on these reported results, the use of Sipuleucel-T has been included in multiple clinical guidelines for the treatment of these patients [63,64,65,66]. In the IMPACT trial, a retrospective analysis of metastatic castration-resistant prostate cancer which had a low baseline PSA level, defined as <22.1 ng/mL, the use of Sipuleucel-T was reported to have an overall survival of 13 months greater than the placebo group [66]. Sipuleucel-T stimulates an antitumour immunological response. Autologous antigen-presenting cells are obtained from specific prostate cancer patients using leukopheresis; the mononuclear cells obtained are incubated with a recombinant protein for 36–44 h. This recombinant protein PA2024 is comprised of two components: firstly, prostatic acid phosphatase (PAP), which is expressed in most prostate carcinomas and at very low levels in other tissues. Secondly, PAP is conjugated with granulocyte–macrophage stimulating factor (GM-CSF) [67]. This conjugate increases the antigen uptake by antigen-presenting cells [67]. This infusion of Sipuleucel-T contains not only cytotoxic T-cells, but also CD4 T-cells, NK cells, antigen-presenting cells and B-cells. Upregulation of serial CD54 measurements is used to monitor the activation of antigen-presenting cells. The first infusion acts as a primer and the second and third as a boost. This increase in CD54 activation was associated with an increased overall survival [68]. In a subgroup of patients from the IMPACT trial, the immune response to Sipuleucin-T was maintained 26 weeks after its administration [67,69].

Fong et al. [70] published a multicentre phase II study of using Sipuleucel-T as neoadjuvant treatment prior to radical prostatectomy for localised prostate cancer. This included a control group who did not receive Sipuleucel-T. From the biopsy and radical prostatectomy samples, immunocytological staining assessed the type and distribution of cells from the immune system. It was reported that in the post-prostatectomy samples patients had increased T-cells at the cancer margin as compared with the biopsy sample, but only those patients who had had prior treatment with Sipuleucel-T, which increased the number of CD4^+^ FOXP3—helper T-cells and cytotoxic T-cells. Although Tregs were also increased, they only formed a small fraction of the total of recruited T-cells. There was no effect on NK-cell populations as would be expected. Approximately 50% of the CD3 T-cells expressed PD-1 and expressed Ki-67 with the implication that they were not exhausted T-cells. This reported result posed the question of whether, combining with other therapies which can increase the immune response of T-cells, such as ipilimumab, or with therapies which target PD-1/PD-L1 may increase the response rates. In the case of metastatic disease, the TME is different and changes in the T-cell subpopulations may be different. Galen et al. [60] showed that the TME of metastatic lesions were immunologically colder than the primary tumour. The onco-sphere is more immunosuppressive with a lower infiltration by lymphocytes and thus T-cell and NK-cell activation is significantly reduced producing immunotolerance and a decreased effect of immunotherapies [71]. However, combining Sipuleucel-T with other therapies may improve progression-free survival with less toxicity. Trials involving the combination with atezolizumab or with radium-223 are ongoing [61,72].

G-VAX is a vaccine using genetically modified irradiated prostate cancer cells using two cell lines, one hormone sensitive and the other hormone resistant. These modified cancer cells express GM-CSF. Theoretically this type of vaccine should increase the differentiation of antigen-presenting dendritic cells [71]. However, the results in asymptomatic and symptomatic patients with metastatic castration-resistant prostate cancer did not show an improved overall survival when compared with G-VAX plus docetaxel, or docetaxel alone. Both the VITAL 1 and VITAL 2 trials were halted before completion due to an increased death rate of patients treated with G-VAX and no increase in the overall survival rates [73].

PROSTVAC-VF is a recombinant vaccine based on a poxvirus which has been modified with a PSA transgene to improve immune-stimulation. It also contains molecules that increase the immune response, such as lymphocyte function-associated antigen 3 (CD58), CD80 and the intracellular adhesion molecule, ICAM-1 or CD54 [74]. This combination is supposed to generate a stronger immune response against tumour cells; however, in clinical trials this has not been seen to be the case [75]. As such the use of this vaccine alone has not been demonstrated to be clinically useful. Although in phase II trials promising results were seen, they were not confirmed in phase III trials [76]. However, when used in combination with a taxane the results suggested an improved progression-free survival as compared with the use of a taxane alone [77]. The use of this vaccine with the concurrent use of immune checkpoint inhibitors, monoclonal antibodies that target receptors that are essential for a successful immune response [78], are ongoing. In patients with localised cancer and castration-resistant prostate cancer, this vaccine is being used in combination with nivolumab (NCT02933255) and in the hormone-sensitive prostate with either nivolumab or ipilimumab (NCT035632217).

**Checkpoint inhibitors** are monoclonal antibodies which can target receptors that play an important role in the immune response [79]. Two pathways of these immune checkpoints that have been studied are associated with cytotoxic T-cell-associated protein 4 (CTLA-4) and the programmed cell death protein 1 (PD-1) [79,80,81,82]. CTLA-4 is confined to the T-cell surface membrane; this protein competes with CD28, a costimulatory receptor, to bind to CD80/CD86 ligands. These two ligands are present on the surface membrane of antigen-presenting cells [82]. The result of this CTLA-4-CD80-CD86 complex is a decrease in T-cell proliferation and the production of Inerleukin-2. These two processes are stimulated if CD28 binds to the CD80/CD86 complex [83]. Differing from CTLA-4, the PD-1 protein is found on the surface membrane of activated T-cells and binds to the PD-L1 and PD-L2 receptors. Both PD-L1 and PD-L2 are frequently upregulated on antigen-presenting cells and cancer cells [84,85]. However, there is a decrease in cytokine production and decreased PD-1 positive T-cell proliferation, activity and their survival in the TME as a result of this treatment [86,87]. In patients with aggressive prostate cancer the upregulation of either of these ligands is associated with a worse prognosis [88]. Bypassing the inhibitory effects of these ligands using antibodies which target these ligands may enhance the immune response against cancer cells.

The results of monotherapy with the monoclonal antibody ipilimumab, a CTLA-4 blocker, in two phase III trials have been reported. In the CA184-43 trial ipilimumab together with radiotherapy to one bone metastasis was used in patients following progression after docetaxel and compared with placebo together with radiotherapy [89]. The CA184-095 trial compared ipilimumab versus placebo before treatment with docetaxel, with the hypothesis that docetaxel is immunosuppressive. However, in neither of the two trials was there a benefit of adding ipilimumab to the patients’ treatment. However, in the final analysis there was a significant difference of long-term survivors in the ipilimumab arm when given after docetaxel chemotherapy [87]. It has been reported that docetaxel is able to remodel the TME, causing an increased intra-tumoral infiltration by T-lymphocytes and upregulation of both PD-1 and PD-L1. In men with metastatic castration-resistant prostate cancer, the combined therapy produced a longer progression-free survival than with anti-PD-1 blockage as monotherapy. Thus, combining therapies may produce beneficial results for these patients [88]. However, the combination of ipilimumab together with Sipuleucel-T in patients with metastatic castration-resistant prostate cancer only produced a low rate of responses. A lower frequency of circulating T-lymphocytes expressing CTLA-4 or prior radiotherapy increased the response rate in these patients [89].

The use of anti-PD1 and anti-PD-L1 is limited to clinical trials in men with metastatic castration-resistant prostate cancer. Phase I trials of the use of anti-PD-1 nivolumab in 17 patients [90] failed to show an objective response. This was also true for the anti-PD-L1 inhibitor avelumab [91] in a group of 18 patients. In the KEYNOTE trial phase I the anti-PD-1 antibody pembrolizumab had some success, with a limited response rate of approximately 17% in those patients positive for PD-L1 expression [92]. In patients pre-treated with docetaxel, the response was also poor, depending on the PD-L1 positivity [93]. The KEYNOTE-199 trial equally reported low response rates in patients previously treated with docetaxel, achieving a 3–5% response rate following the use of pembrolizumab [94]. In the KEYNOTE-641 and IMbasador trials combining atezolizumab or pembrolizumab with enzalutamide showed no improvement in terms of overall survival as compared with enzalutamide alone [95,96]. A phase 3 randomised trial conducted in 2017 used ipilimumab monotherapy in high doses; 10 mg/kg was compared with a placebo. There was no difference in the overall survival between the two groups: 28.7 versus 29.7 months, respectively. In terms of progression-free survival, this was reported as 5.6 versus 3.8 months, respectively, and a higher decrease in the levels of PSA. The treatment group had an increased frequency of 3–4 grade adverse effects and an increased death rate in the treatment group. This highlights the difference between a statistically significant difference and a clinically significant difference.

Dostarlimab was approved by the FDA for use in patients with a mismatch repair deficiency (dMMR) in castration-resistant metastatic prostate cancer. It had previously been approved in the treatment of endometrial cancer and locally advanced rectal cancer [97]. The FDA has extended this approval to solid tumours which have dMMR deficiency. However, in metastatic castration-resistant prostate cancer, the presence of dMMR has been reported to be relatively rare, with a prevalence of approximately 1–3% [98]. The implication is that monotherapy in patients harbouring the dMMR mutation is limited by the small number of patients. Its mechanism of action is to block the PD-L1 receptor preventing tumour cells from escaping immune surveillance. Clinical trials using dostarlimab alone or in combination in castration-resistant metastatic prostate cancer are ongoing [99,100].

## 4. Bispecific Antibodies That Target Costimulatory Receptors of T-Cells

The evolving array of precision antitumour immunotherapy capable of targeting two epitopes in a single treatment is evolving and may play a crucial role in anti-cancer therapies [101,102].

These bispecific antibodies incorporate the epitope binding domains of two monoclonal antibodies, firstly, against a specific cancer cell epitope, such as PMSA and a costimulatory T-cell receptor, enabling the T-cell to bind to the tumour cell and thus improving its cytotoxic role [103,104].

These bivalent monoclonal antibodies are obtained against monospecific epitopes or even tri-specific T-cell antibodies (BiTEs and TriTEs) [105]. It is pivotal that the optimisation of the manufacturing process is to achieve the optimal binding of the antibody to the desired epitope, the stability of the molecule and its pharmakinetics. The dual targeting of the cancer cell with the direct contact with the T-cell produces the activation, proliferation and cytotoxicity of these cells which are antigen dependent. Thus, by binding of the antibody to the tumour antigen such as PSMA, the crosslinking between the T-cell and tumour cell improves the cytotoxicity of the T-cell, a type of immunological symbiosis. This results via CD28 or CD3 activate the adjacent T-cells, resulting in an antigen-specific T-cell immune response directly against cells expressing this epitope. These BITEs have been used in pre-clinical trials [106] and have been reported to cause significant antitumoral effects in mouse xenograft models. In these in vivo experiments, there was a dose-dependent inhibition of tumour growth, although in tumours expressing a lower level of the selected epitope the therapy had a decreased effect [107]. Figure 2 shows the action of BiTEs with T-cell activation which eliminates the tumour cells.

Newer BiTEs have a longer half-life, such as AMG160 which by binding to the PMSA epitope induces T-cell activation and the elimination of tumour cells [108]. At the ESMO 2020 congress, a Phase 1 clinical trial reported a dose-dependent PSA reduction in 68% of patients and a reduction of >50% in 34% of participating patients. However, some 26% of patients experienced the cytokine release syndrome, which was most severe during the first cycle of treatment [109]. The use of treatments with BiTEs and the cytokine release syndrome has been recently reviewed [110]. It is a systemic inflammatory response with a large rapid release of cytokines into the circulation by the immunological system and may result in multi-organ failure, with lung, liver and/or kidney failure, and ultimately lead to the death of the patient. This has been reported in treatments with immune checkpoint inhibitors, BiTEs and more frequently with CAR-T cells. A multicentre phase I study of AMG340 an anti-PMA-CD3 BiTE is ongoing as well as the anti-PMSA-CD28 plus the anti-PD1 antibody cemiplimab.

More recently, tarlatamab is a new class of these drugs, with an extended half-life, and it is a BiTE that targets the delta-like ligand 3 (DLL-3) with a CD3 T-cell engager which has been developed for the treatment of neuroendocrine prostate cancer. It binds to the tumour cell surface DLL3 and CD3 on cytotoxic T-cells, resulting in their activation, the release of cytokines and death of DLL-3 tumour cells [111].

Other potential tumour epitopes such as Glypican-1, disintegrin and metalloproteinase 17 and anti-prostate cancer stem cells are being developed [111]. Other developments include the use of subcutaneous treatments and the use of injectable biopolymer depots to provide a more sustained release of the BiTE [111].

Frequently prostate cancer patients are elderly and may have co-existing cardiovascular disease. The use of both PD-1 and CTLA-4 inhibitors can cause cardiovascular adverse effects. In a meta-analysis of 5463 patients, a total of 174 patients suffered cardiotoxicity, or 3%, atrial fibrillation being the most common adverse event (12%), cardiac failure (6%) and death in three cases [111]. Thus, treating physicians should be aware of these as cardiovascular adverse effects are underestimated [111]

## 5. Chimeric Antigen Receptor (CAR) T-Cell Therapy

The patients’ T-cells are removed using leukopheresis and are modified genetically to express synthetic receptors directed against tumour-specific epitopes. After undergoing culture to expand their numbers, they are re-infused into the patient where in an MHC-independent mode they can potentially eliminate the tumour cells [112]. Second and third generation CATs contain intracellular costimulatory domains to further enhance both the activation and proliferation of CAR-T cells [113,114]. Third-generation CAR-T cells appear to have an increased activity against cancer cells as compared to second generation CAR-T cells, at least in pre-clinical trials [115].

The heterogeneity of tumour epitopes makes the design of CAR-T cells more difficult; plus, the fact that these epitopes must not be or must be very lowly expressed in normal cells to prevent tissue damage. PMSA and prostate stem cell antigen have been used along with co-stimulants such as tumour growth factor-β to overcome the immunosuppressive TME [116]. Synergy with the addition of zoledronate was seen when gamma-delta CAR-T cells were used to target prostate cancer cells, the CAR-T cells recognising the phosphor-antigens in the TME [117]. It has been reported that CAR-T cells which are engineered to express cytokines, IL-12, IL15 or CCL19 give the CAR-T cells a higher proliferation rate, persistence in the animal model and higher antitumoral capacities than conventional CAR-T cells [117].

Pre-clinical studies using mice xenograft models to determine dosage and toxicity are ongoing. To improve the effectiveness of CAR-T therapy chemotherapy to cause lymphodepletion has been investigated, normally using cyclophosphamide and/or fludarabine infusions for three days, starting five days before the infusion of the CAR-T cells. The hypothesis of this Phase 1 trial is that pre-treatment with chemotherapy may decrease the number of Tregs and there would be less inhibition of the CAR-modified T-cells [118].

However, CAR-T therapy has its side effects: allergic reactions, septicaemia for the immunosuppression, cytopenia, the immune effector cell causing neurotoxicity and the cytokine release syndrome. These last two adverse effects were thought to be the principal reasons for the death of patients [119]. In addition, if the cancer cells do not express the targeted epitope or the immunosuppressive effect is elevated the efficacy of CART treatment may be limited.

## 6. PMSA-Linked Radionuclides

The expression of the PMSA epitope is increased in approximately 75–95% of patients with metastatic castration-resistant prostate cancer. PMSA antibodies linked to a beta-emitting radionuclide such as lutetium 177 (^177^Lu) and yttrium 90 (^90^Y) have been shown in phase I and phase II trials to have activity against PMSA-expressing cancer cells as well as a dose-dependent response [120,121]. The PSMA epitope is expressed in nearly all these cancer cells; however, the use of PSMA imaging by PSMA PET-CAT or whole-body single-photon gamma camera computerised tomography has not been shown to be useful in detecting PMSA-expressing tumours. The use of this therapy has been based on the response seen by imaging studies. It has been hypothesised that the higher the expression of PMSA on the cancer cells, the better the response. With higher doses of ^177^Lu, even patients with a negative imaging study responded. The authors suggested that micro-metastatic disease was eliminated or decreased and was too small to be detected by imaging studies [122]. No differences in progression-free survival were detected between patients treated with cabazitaxel or ^177^Lu-PMSA in patients that progressed after docetaxel chemotherapy [123]. It is interesting that the authors mention micro-metastatic disease, as the 2022 US Food and Drug Administration [124] and European Medicines Agency [125] approved its use as a last-line therapy after progression of the disease following androgen deprivation and taxane-based hemotherapy. After the anti-PSMA antibody binds to the cancer cell, the radionuclide enters the cell, dispersing throughout the cytoplasm. The beta irradiation by damaging the DNA leads to the elimination of the cancer cell. The low penetration of these radioactive particle’s limits damage to the neighbouring normal tissues [126]. Studies on improving this type of therapy are ongoing [127].

## 7. Androgen Deprivation Therapy

Since the pioneering work of Huggins and Hodges, where in 1941 they showed that surgical castration prolonged progression-free survival and overall survival in men with metastatic prostate cancer, androgen deprivation has been the cornerstone of prostate cancer treatment [128]. The development and use of steroidal (megestrol, cypertone and medroxyprogestone) plus androgen receptor antagonists such as flutamide and bicalutamide were developed in the 1960s and 70s. These effective androgen receptor blockers inhibit the binding of testosterone and dihydrotestosterone (DHT) to the androgen receptor. However, their usage prolongs the progression-free survival for approximately 3–5 years, whereby the selection of resistant cells and changes in the androgen receptor leads to resistant tumour cells. With the development of GnRH agonists such as leuprolide and goserelin and GnLH antagonists such as degarelix, which is injectable, or the newer oral relugolix, these treatments are often used in combination with androgen receptor blockages. The newer drugs such as abiterone and enzutamide are normally used as second-line therapies, although the new NCCN guidelines [63] suggest that they may stop being used as a first-line treatment. After progression, chemotherapy with or without androgen deprivation therapy is recommended. Immunotherapy remains as a fourth-line therapy. The important question is, if it is known that androgen receptor blockades in their many forms eventually cause castration-resistant prostate cancer, why is it used as a frontline therapy? [129].

## 8. Mechanisms of Resistance to Immunotherapy

Resistance to immunotherapy occurs with eventual progressive disease; there are different mechanisms for this occurrence. As immunotherapy in castration-resistant metastatic prostate cancer patients is a fourth-line therapy, the results of improvement in both progression-free survival and overall survival are limited. In metastatic prostate cancer, lesions can be measured according to the RECIST criteria as well as serum PSA levels and the results of immunotherapy determined. There are two main mechanisms for the failure of immunotherapy; one is physical and the other is immunological.

**Physical barriers for the failure of immunotherapy**: As mentioned previously, this cancer is formed by many different cell types, one of which is cancer-associated fibroblasts. These cells produce stiffening in the extracellular matrix in addition to producing collagen, proteins, glyco-proteins and proteoglycans, which form aggregates such as sheet-line networks and fibrils. In addition, it has biophysical properties such as molecular density, rigidity and tensional forces [42]. This barrier prevents the ingress of cell-based therapies such as CAR-T and BiTES and of larger molecules, thus reducing their effectiveness. Not only that, but because of the size of the tumour metastasis, immunotherapy may not be able to eradicate all the tumour cells.

**Immunological mechanisms of resistance to immunotherapy:** Immunotherapy targets specific epitopes such as PMSA or by a diminution of the cytotoxic immune defences. Immunotherapy resistance has been defined as primary or secondary [130]. Primary resistance has been defined as that which occurs at the onset of treatment with immunotherapy, whereas acquired or secondary resistance develops overtime with treatment [130]. This is due to heterogeneity of prostate cancer cells [6,27], which is manifested through genetic alterations and dysregulation of the host’s immunological system. This combined complex interplay of genetics and the microenvironment may result in the resistance to immunotherapy [131]. The efficiency of immunotherapy can be affected by genetic and molecular alterations in the primary tumour, such as the mitogen-activated protein kinase (MAPK), which upon activation leads to secretion of cytokines such as VEGF and IL-10, which decreases cytotoxic T-cell effectiveness and infiltration of the tumour. This results in a decrease in the effectiveness of T-cell mediated immunotherapy [132]. The loss of the tumour suppressor gene PTEN is associated with a reduced immunological response and fewer cytotoxic T-cells. PTEN mutations have been described in 20% of primary prostate cancers and in 50% of castration-resistant tumours [133], with a higher rate of resistance to immune checkpoint inhibitors [134]. The activation of the WNT signalling pathway decreases the activity of cytotoxic lymphocytes due to the stabilisation of ß-catenin, which in prostate cancer has been linked to both tumour progression and resistance to treatment [135]. In murine models, an increased expression of ß-catenin decreases the number of dendritic cells, which leads to fewer cytotoxic T-lymphocytes and decreases the effectiveness of immune checkpoint inhibitors [136].

In prostate cancer, the loss of ß2-microglobulin decreases the expression of MHC-I molecules on the cells’ surface, thus hindering cytotoxic T-cells from recognising tumour cells and facilitating resistance to immune checkpoint inhibitors as well as T-cell based immunotherapy [137], resistance to PD-1 inhibitors and increased levels of cancer-associated fibroblasts [138]. 

The synergistic activity of vascular endothelial growth factor with IL-8 creates an immunosuppressive TME, favouring tumour growth and less accessibility of the ingress of immune effector cells to enter the tumour and thus eliminate tumour cells [139]. The increased level of IL-8 also recruits MDSCs, CD15^+^ monocytes and neutrophils and thus increases the suppression of adaptive T-cell antitumour immunity and the efficacy of immune checkpoint inhibitors [140]. Continuous exposure of the TME to interferon gamma leads to genetic instability of cancer cells and later, by an alteration in DNA repair and editing genes, results in the development of resistance to immunotherapy [141]. This causes specific mutations in the JAK-1 and JAK-2 signalling pathways and is associated with resistance to anti-PD-1 and CTLA-4 therapies. Cancer cells with these mutations often lack the expression of PD-L1 [142]. The TME in the context of immunotherapy decreases anti-PD-1 and PD-L1 immune checkpoint inhibitors by increasing the numbers of Tregs and tumour-associated macrophages [143]. Similarly, TGF-beta has been associated with a decreased efficacy of anti-PD1 immunotherapy, increasing cancer-associated fibroblasts and as previously mentioned, this leads to a decreased infiltration of the tumour by cytotoxic T-cells [144]. There is also acquired resistance to immunotherapy; epigenetic modifications have a role to play against immune checkpoint inhibitors but also may increase the sensibility to these therapies, attracting cytotoxic T-cells, at least in lung cancer [145]. Irrespective of the tumoral mutational burden, the genetic and epigenetic changes that decrease tumour antigens are associated with both primary and acquired resistance to immune checkpoint inhibitors [146]. As mentioned previously, MDSCs create an immunosuppressive environment not only with regards to Tregs and M2 macrophages but also diminished responses to immune checkpoint inhibitors; this includes the participation of other myeloid cells such as tumour-associated macrophages. These cells are enriched in castration-resistant metastatic prostate cancer and as such decrease cytotoxic T-cell activity, promoting resistance to immune checkpoint inhibitors. The use of adenosine A2A inhibitors reverses the immunosuppressive effect on cytotoxic T-cells and increases the response to PD-1 blockage. Inhibition of A2ARs using ciforadenant combined with atezolizumab induced clinical responses in castration-resistant metastatic prostate cancer [147]. In prostate cancer, a significant proportion of the tumoral mass consists of TAMs with an M2 phenotype, which release cytokines that suppress cytotoxic T-cells, recruit Tregs and express PD-L1, which results in a decreased effect of PD-L1 blockade [148]. Platelets play a fundamental role in the TME; in colon cancer, platelets that are positive for the expression of PD-L1 are associated with a poorer prognosis and a decreased progression-free survival. This transfer of PD-L1 from cancer cells to the platelets in the TME depends upon direct cell-to-cell contact and, vice versa, platelets can induce the expression of PD-L1 expression in the tumour cells and are associated with cytotoxic T-cell exhaustion [149].

In summary, after the use of androgen blockage and taxane-based chemotherapy, the use of immunotherapy has a limited increase in progression-free survival and overall survival. Combined immunotherapy may improve progression-free and overall survival. The phase II INSPIRE trial using the combination of ipilimumab with nivolumab showed a benefit in progression-free survival as compared with patients naïve to immune checkpoint inhibitors.

## 9. Immunotherapy in the Context of Previous Knowledge

For over eight decades, the mainstay of the treatment of metastatic prostate cancer was surgical castration, which increased progression-free survival and overall survival by some three to five years before progression [128]. In the 1960s and 1970s, oral androgen receptor antagonists were developed, such as flutamide and bicalutamide, to treat metastatic prostate cancer. These drugs have a similar progression to surgical castration. In the era of PSA screening, failure of curative treatment is detected before the appearance of macro-metastasis. Similarly, the use of abiraterone or enzalutamide prolongs progression-free survival. 

However, there are drawbacks when using ADT: firstly ADT-resistant cells are being selected and thus survive and proliferate, leading to the eventual development of macro-metastasis. Secondly, they have systemic side effects in an otherwise asymptomatic patient: skeletal complications, metabolic complications such as glucose and lipid levels, cardiovascular complications such as sexual dysfunction as well as cognitive and mood disorders [150]. These complications are important as they may result in increased morbidity and mortality rates [150]. Thirdly, ADT affects the immune system; the use of ADT with degarelix increased the number of cytotoxic T-cells but also more significantly the number of Tregs [151,152]. ADT has also been reported to impair the adaptive immune responses, increasing the immunosuppressive environment of the TME [153]. In other studies, cytotoxic T-cells were increased, enhancing PD-1 therapy [154]. ADT also suppresses the antitumour neutrophil efficacy; progression of prostate cancer is associated with decreased cytotoxicity caused by neutrophils, especially with second-generation ADT [155]. Enzalutamide-induced resistance activates immunosuppressive macrophages, which are a major component of bone metastatic prostate cancer [156]. Resistance to ADT is usually followed by chemotherapy; the taxanes cause transient immunomodulation, improving progression-free survival; however, they have their systemic adverse effects, including fatigue, myalgia, arthralgia, neutropenia and peripheral neuropathy [157].

Over the last eight decades, the treatment of metastatic prostate cancer and later biochemical failure has radically changed. With the development of new technologies, such as CTC detection, circulating tumour DNA and next-generation sequencing, these older studies have laid the groundwork for a better understanding of the behaviour of prostate cancer, its eradication or progression to metastatic disease. The importance of the immune system in the pathological behaviour of prostate cancer and the dynamic interplay between tumour cells, immune cells, both effector and suppressive, as well as the effect of the TME has been studied intensively. An idea therapy would be specific to prostate cells with the minimum of toxicity. Thus, the manipulation of the immune system to a less immunosuppressive one, which is a dynamic process for each onco-sphere, may herald new breakthroughs in the treatment of prostate cancer. The earlier publications have formed the foundation for new therapies for the treatment of prostate cancer. The discovery that the PMSA antigen is expressed only on prostate cells and in most tumour cells is leading to the development of PMSA-directed therapy such as BiTEs, radioligand therapy and the elucidation of signalling pathways, and the development of checkpoint inhibitors may be beneficial to patients with prostate cancer.

## 10. Should Immunotherapy Be Used as Frontline Treatment for Biochemical Relapse? Are We Not Seeing the Wood for the Trees?

Immunotherapy is used after the failure of ADT and taxane-based chemotherapy. As a third- or fourth-line therapy for castration-resistant metastatic prostate cancer, the results have not been promising and much less effective than in other solid tumours [61]. 

The use of monotherapy with nivolumab, avelumab and pembrolizumab had limited success, while the combination of an anti-PD-1 plus enzalutamide showed no survival improvements [158]. The IMPACT trial using Sipuleucel-T as monotherapy demonstrated that patients with low baseline PSA values had a better response than those patients with adverse prognostic factors and suggested that early immunotherapy as a treatment strategy may be more beneficial to patients with castration-resistant metastatic disease. and thus, patients with a low tumour burden may benefit from Sipuleucel-T therapy [159].

Combined immunotherapy may improve progression-free status and overall survival. However, the phase II INSPIRE trail using the combination of ipilimumab with nivolumab showed a benefit to progression-free survival as compared with patients naïve to immune checkpoint inhibitors, especially in patients with the dMMR mutation [158]. Despite the use of biomarkers for mutations in DNA repair genes, phosphatase and tensin homologues and homologous recombination repair genes to choose the best treatment for these patients, only Olaparib in the PROFOUND study and Lutetium-77 in the Vision trial showed an increased survival rate [160,161]. The small subset of metastatic castration-resistant prostate cancer, comprising 3–5% of these tumours, may benefit from combined therapy. Those patients with high microsatellite instability, mismatch repair deficiency or elevated tumour mutation burden may benefit from single-agent immune checkpoint inhibitors [162]. A case report showed that a patient who progressed following treatment with pembrolizumab and then treated with a PSMA-CD3 bispecific T-cell antibody and then restarted with pembrolizumab at progression showed an undetectable PSA level for over 11 months; thus, bispecific antibodies may cause re-sensitisation to immune checkpoint inhibitors However, combination therapy such as pembrolizumab combined with chemotherapy, next-generation hormonal agents or PARP inhibitors in the KEYNOTE 641, 921, 991 and KEYLYNK 010 showed no benefit in unselected patients with metastatic castration-resistant prostate cancer [163]. The use of abiraterone, Olaparib or the combination of both drugs as first-line treatment in metastatic castration-resistant prostate cancer with DNA repair defects showed a benefit from combined therapy versus monotherapy [164]. This suggests that genetic testing in these patients may determine the best treatment options. However, the KEYNOTE-010 trial on heavily pre-treated metastatic castration-resistant prostate patients who had previously failed treatment with abiraterone or enzalutamide, and a taxane comparing pembrolizumab plus Olaparib versus abiraterone or enzalutamide, showed no significant survival differences in unselected patients [163]. To highlight the need for genetic testing, approximately 20–25% of patients have alterations in the genotype, especially BRCA mutations. Although these patients have a more aggressive disease, there is an increased sensitivity to PARP inhibitors. The PROFOUND trail compared Olaparib versus abiraterone or enzalutamide in patients with BRCA mutations. Olaparib proved to be superior in these patients who had been previously treated with either abiraterone or enzalutamide [164]. In the PROpel study, Olaparib plus abiraterone versus placebo with abiraterone as first-line treatment showed improved overall survival in unselected patients with metastatic castration-resistant prostate cancer who had previously been treated with taxane-based chemotherapy [162].

With the limited success of immunotherapy in the setting of metastatic castration-resistant prostate cancer, its use as a neoadjuvant therapy or at the time of biochemical failure would seem to be a treatment option. As mentioned in the section Mechanisms of Resistance to Immunotherapy, it is suggested that micro-metastatic disease may be more sensitive to immunotherapy. There are reports of the use immunotherapy in these patients but large-scale, randomised double-blinded trials have not been performed. Madan et al. [165] reported the use of flutamide alone versus flutamide plus the vaccine PROSTVAC, but failed to report a benefit of adding the vaccine to flutamide. In a single-arm phase I/II trail (EudraCT 2009-017259-91), the use of autologous dendritic cells reported an improved PSA doubling time 18.9 months versus 5.7 months in patients who did not receive treatment. After a second cycle of dendritic cell infusions, the PSA doubling time increased to 58 months. The authors suggested that long-term immunotherapy in patients with early signs of biochemical failure led to a significant prolongation of the PSA doubling time and avoided the use of androgen deprivation therapy [166]. In another Phase I/II trial of dendritic cell vaccines after being treated with prostatectomy for high-risk prostate cancer, the authors reported that all patients who experienced biochemical failure had a stable disease for a median of 99 months. Those who had extra-prostatic extension of their cancer and responded to the vaccine, the time to biochemical failure was longer than in non-responders [167]. A human phase I study of PF-06753512, a vaccine-based immunotherapy regimen (VBIR) in non-metastatic hormone sensitive biochemical recurrence and metastatic castration-resistant prostate cancer (mCRPC), was carried out. In patients with biochemical failure, the response rate was 25% and the duration of PSA doubling time was 10 months after the vaccine plus sasanlimab [168].

Neoadjuvant immunotherapy has also been used before definitive curative therapy; the use of the Sipuleucel-T vaccine before radical prostatectomy increased immune effector cells as compared with the control group [169]. High-risk prostate cancer has a high frequency of biochemical failure, metastasis and death after curative treatment. This recurrence is thought to be the result of micro-metastasis not detected by conventional imaging. Although the study did not give details of the rates of biochemical failure, the authors suggested that the treatment could cause cell death and an antitumour systemic immune response. Disseminated tumour cells detected in bone marrow aspirates act as a reservoir for future metastatic progress [170] (Figure 3).

Surprisingly, the use of rituximab, an anti-CD20 monoclonal antibody targeting B-cell lymphocytes, as neoadjuvant therapy prior to radical prostatectomy not only reduced intra-tumoral B-cells but also CD3-positive T-cell lymphocytes. The results of B-cell lymphocytes and biochemical failure with the use of rituximab have been conflicting [171,172].

If the PSA levels are low or there is no evidence of macro-metastasis on imaging studies, is it possible to detect changes following immunotherapy and determine which type of immunotherapy maybe the best option. With advancing technology, the detection of circulating, tumour cells (CTCs) maybe the answer to this question (Figure 4). The only FDA-approved method for the detection of CTCs is in patients with metastatic prostate cancer (CellSearch^®^ system, Merarin Silicon Biosystems, San Diego, CA, USA), which detects EpCAM-positive cells in the circulation. This method cannot detect prostate cells which have undergone the epithelial mesenchymal transition where epithelial markers are not expressed. However, a more recent study suggested that PMSA expression in CTCs is complementary to a PET scan. Those cells which did not express PMSA had a poorer response to PMSA-targeted therapy [173]. Furthermore, using differing immunocytochemical methods, PD-L1, CTLA-4 and androgen receptor variants can be detected to select the best option of immunotherapy as well as monitoring the response to treatment [174,175]. Zhang et al. [175] reported that the expression of immune checkpoints on circulating tumour cells in men with metastatic prostate cancer may be able to monitor the results of immunotherapy.

## 11. Conclusions: Future Developments and the Use of Immunotherapy in Patients with Prostate Cancer

The development of new agents using pre-clinical and Phase I and II clinical trials may decrease the risk of developing metastatic disease or prolong the dormancy or latent period before there are changes in the clonal instability of the tumour cells or changes in the immunological response. Although there is evidence that the early use of immunotherapy may increase the time to biochemical failure, double-blind, randomised placebo controlled multicentre trials need to be conducted to find when it is most appropriate to use immunotherapy and what immunotherapy is the best option for patients treated with curative therapy for prostate cancer. There are still important questions to be answered. When should immunotherapy be administered? Should it be combined immunotherapy, or combined with ADT, chemotherapy or radiotherapy? In which sequence should therapies be administered? As well as what therapies also affect normal tissues and the side-effect profile.

## Figures and Tables

**Figure 1 biomedicines-13-01179-f001:**
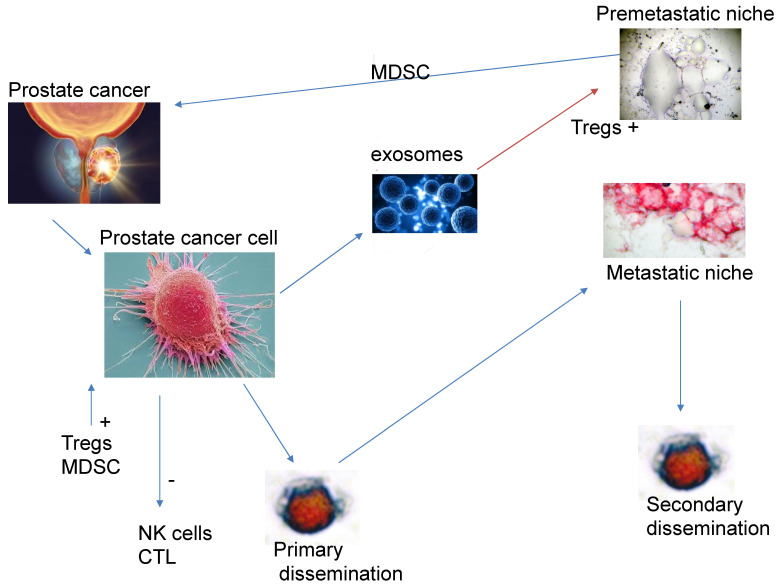
Effects of prostate cancer on the immune system.

**Figure 2 biomedicines-13-01179-f002:**
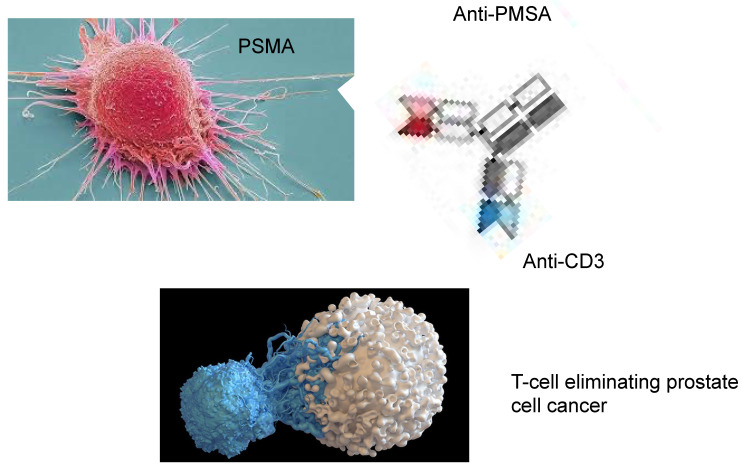
The action of a BiTE expressing bivalent antibodies against PMSA and permitting the binding to T-cell CD3 epitope.

**Figure 3 biomedicines-13-01179-f003:**
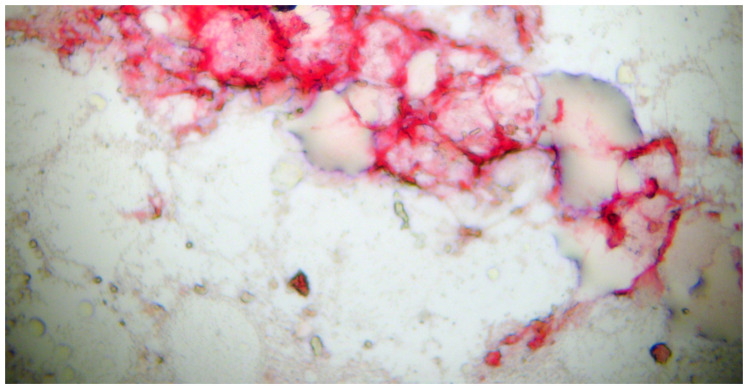
Bone marrow infiltration with PSA tumour cells (staining red).

**Figure 4 biomedicines-13-01179-f004:**
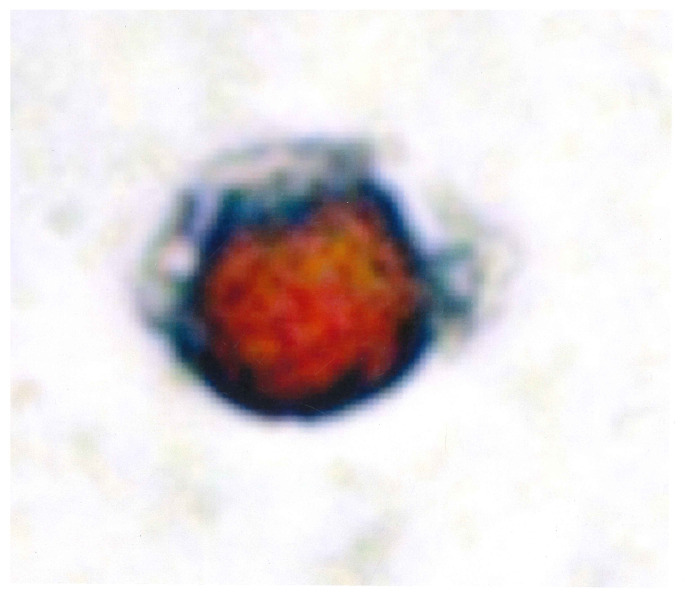
Circulating tumour cell-expressing PSA (red) and MMP-2 (black).

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
