# Peer review of "Immunomodulation and Immunotherapy for Patients with Prostate Cancer: An Up-to-Date Review"

_biomedicines, 2025, doi:10.3390/biomedicines13051179_

Round 1
Reviewer 1 Report
Comments and Suggestions for Authors
Dear Author
Thanks for inviting me the review the manuscript “Immunomodulation and immunotherapy for patients with 2 prostate cancer: an up-to-date review”. I can say the immunolmodulation is explained in general and more specific prostate cancer immunology should be added to the manuscript.
Line 74: prostate cancer cells: Tumor cells can directly affect the immune system both. Please explain almost all prostate cancers are adenocarcinomas. These cancers develop from the gland cells in the prostate (the cells that make the prostate fluid that is added to the semen). Other types of cancer that can start in the prostate include: Small cell carcinoma (small cell neuroendocrine carcinoma).
Based on the MD Anderson reports the https://www.cancerresearch.org/cancer-types/prostate-cancer
Immunotherapy for prostate cancer includes two FDA-approved treatment options and is a promising area of research for metastatic cancer treatment.
- Sipuleucel-T (Provenge®): a vaccine composed of patients’ own immune cells, which have been stimulated to target the PAP (prostatic acid phosphatase) protein highly expressed on prostate cancers; approved for subsets of patients with advanced prostate cancer
Immunomodulatory
- Dostarlimab (Jemperli): a checkpoint inhibitor that targets the PD-1/PD-L1 pathway; approved for subsets of patients with advanced prostate cancer that has DNA mismatch repair deficiency (dMMR)
- Pembrolizumab (Keytruda®): a checkpoint inhibitor that targets the PD-1/PD-L1 pathway; approved for subsets of patients with advanced prostate cancer that has high microsatellite instability (MSI-H), DNA mismatch repair deficiency (dMMR), or high tumor mutational burden (TMB-H)
In the manuscript I expected to explain these two immunomodulatory
What is the conclusion of the manuscript?
Author Response
Reviewer 1: Many thanks for reviewing this article and your comments to improve it. The changes to your suggestions are in blue
- Lines 28-96 cover the morespecific details of prostate cancer immunotherapy
- Lines 351-361 cover the new FDA approved drugsfor the treatment of metastatic castration resistnat prostate cancer and explain the immunomodulatory mechanisms of these treatments
- The conclusión of the manuscript has been extended

Reviewer 2 Report
Comments and Suggestions for Authors
The author in this review article have attempted to summarize the current state of treating prostate cancers using immunotherapy. Though the attempt is appreciated, the information has extreme concurrence in the information compiled in other recently published review papers in the same topic such as PMID: 37762648, PMID: 37366915, PMID: 36925917, PMID: 37738978. This manuscript has very less information to add to the existing literature and does not provide a critical perspective. The manuscript also needs a thorough english language correction. The authors are advised to reframe the content focusing on adding information on the probable resistance mechanisms and improve the quality of the figures to enhance the importance of the work.
Comments on the Quality of English LanguageThe manuscript also needs a thorough English language correction.
Author Response
Reviewer 2: Many thanks for reviewing this article and your comments to improve it. The changes to your suggestions are in red
- With respect to your comments about the 2023 reviews, this new review of the 178 references, 22 (12%) are from 2023; 17 (10%) from 2024 and 9 (5%) from 2025 or in other words 27% of the references refer to new material. As for the more older references I feel it is essential to put the new data in context with what is already known.
- The English has been corrected and errors eliminated
- There is a new section of mechanisms of resistance to immunotherapy lines 494-579. And also on immunotherapy lines 584-679

Round 2
Reviewer 2 Report
Comments and Suggestions for Authors
The author has added discussions on the previously suggested sections.
Comments on the Quality of English LanguageThere are a few english language errors that needs to be fixed.
Author Response
Thank you once again to the reviewers for their work
- The English has been revised and corrected
- A new section from lines 580, page 13 to line 623 page 14 has been included to address the progress in cancer treatment based on earlier work that has led to a better understanding of prostate cancer progression. I think it is important to put the new findings post 2022 in context with earlier reports dating back to the 1940s. These new findings have led from empirical treatments such as surgical or medical castration to more specific therapies. Prostate cancer and its progression has multiple players in its onco-sphere, understanding the immunological factors and signalling pathways may help to develop new less toxic therapies to improve the prognosis in patients with prostate cancer.
